# Ergonomic Factors That Impact Job Satisfaction and Occupational Health during the SARS-CoV-2 Pandemic Based on a Structural Equation Model: A Cross-Sectional Exploratory Analysis of University Workers

**DOI:** 10.3390/ijerph191710714

**Published:** 2022-08-28

**Authors:** Víctor Manuel Ramos-García, Josué Aarón López-Leyva, Raúl Ignacio Ramos-García, Juan José García-Ochoa, Iván Ochoa-Vázquez, Paulina Guerrero-Ortega, Rafael Verdugo-Miranda, Saúl Verdugo-Miranda

**Affiliations:** 1Departamento de Física, Matemáticas e Ingeniería, Universidad de Sonora, Navojoa 85880, Mexico; 2Centro de Innovación y Diseño, CETyS Universidad, Ensenada 22860, Mexico; 3Independent Researcher, Long Beach, CA 90813, USA; 4Academia de Matemáticas, Centro de Bachillerato Tecnológico Industrial y de Servicios No. 64, Navojoa 85860, Mexico

**Keywords:** structural equation model, job satisfaction, organizational ergonomics, physical ergonomics

## Abstract

This paper presents a structural equation model to determine the job satisfaction and occupational health impacts concerning organizational and physical ergonomics, using (as a study) objective unionized workers from the University of Sonora, South Campus, as an educational enterprise, during the SARS-CoV-2 pandemic. The above is a key element of an organizational sustainability framework. In fact, there exists a knowledge gap about the relationship between diverse ergonomic factors, job satisfaction, and occupational health, in the educational institution’s context. The method used was a stratified sample of workers to which a job satisfaction–occupational health questionnaire was applied, consisting of 31 items with three-dimensional variables. As a result, the overall Cronbach’s Alpha coefficient was determined, 0.9028, which is considered adequate to guarantee reliability (i.e., very high magnitude). Therefore, after the structural equation model, only 12 items presented a strong correlation, with a good model fit of 0.036 based on the root mean square error of approximation, 1.09 degrees of freedom for the chi-square, 0.9 for the goodness of fit index, and a confidence level of 95%. Organizational and physical factors have positive impacts on job satisfaction with factor loads of 0.37 and 0.53, respectively, and *p*-values of 0.016 and 0.000, respectively. The constructs related to occupational health that are considered less important by the workers were also determined, which would imply a mitigation strategy. The results contribute to the body of knowledge concerning the ergonomic dimensions mentioned and support organizational sustainability improvements in educational institutions and other sectors.

## 1. Introduction

Since the 1930s, there has been great interest in job satisfaction research, which probably peaked in the 1960s. In the 1980s, this issue began to be more closely related to the quality of life at work, its impact on mental health, and the relationships between coworkers and family, with a growing concern for the individual’s personal development in an educational context throughout life [1]. Nowadays, job satisfaction is one of the most important issues according to the subjective perception of the worker. Therefore, researchers have focused on developing studies of this nature. In fact, human resources are of great value within organizations; the impact can be measured through the business transformation, where the success of the company is guaranteed through experience, knowledge, motivation, and the management of labor environment changes [2]. Hence, it makes it possible for organizations to achieve their business objectives and results through their employees, based on the relevance of seeing workers as valuable capital, which determines an important weight in the achievement of the objectives to follow [3]. Thus, it has been found that job satisfaction is an important and highly beneficial element for both organizations and employees in an organizational sustainability framework [4,5]. Generally, this element is a pleasant or positive emotional state of subjective perception, resulting from the evaluation of work experiences, and it heavily impacts the behavior of the worker [4,6,7]. In particular, organizational sustainability can be understood as the organization’s capacity based on its internal and external processes in order to maintain and improve its operation mode to increase performance, profitability, and competitiveness.

Around the world, interest in the work environment in institutions/organizations has gained enormous relevance due to the various problems faced, largely due to internal problems [8]. Inside them, one of the main issues is the lack of job satisfaction, which inhibits the development of creative and innovative work [9]. In consequence, job dissatisfaction affects the worker’s performance as well as productivity, causing demotivation and a lack of interest in his/her work, which produces worker apathy and possibly implies that the worker is not correctly complying with his/her daily or habitual assigned functions [10]. On the other hand, this situation can lead to anxiety or stress and, in extreme cases, the worker can suffer from depression [11]. Universities (or academic institutions) are no exception, although it is true that they generate professionals in specialized areas, such as manufacturing, engineering, and services, among others, they must also improve their internal production processes in the workplace as part of an organizational sustainability framework, supporting the regional sustainability development [12].

Furthermore, the factors that influence job satisfaction are essential to improve the well-being of (a large part of) society and high-performance jobs [5,13]. At the same time, a satisfaction index can be established based on the working conditions, allowing one to determine the main deficient elements in which action must be taken to achieve improvements in the work environment. In addition, these factors constitute great important elements for the development of all processes where human resources intercede. Therefore, the measurement of these elements is important to determine the relationships of the factors that most affect and impact the job satisfaction of the study object. The literature shows that the study of job satisfaction has been approached from multiple dimensions, and is linked to some variables, such as motivation, sociocultural aspects, and economic and communication features, among others [10]. However, it has not been addressed with ergonomic approaches in higher-level institutions focused on staff who are unionized and provide operational services (maintenance, security, infrastructure hygiene, drives, and others) because this scenario is not widely studied, i.e., case studies and related information are scarce [14,15]. Therefore, this study aims to determine the impacts of ergonomic factors on job satisfaction and occupational health concerning unionized workers who provide operational services at an institution through structural equation analysis.

The rest of this manuscript is organized as follows: In Section 2, the contextual and theoretical backgrounds are explained. In particular, Section 2.1 describes the dimensions of ergonomics. Section 2.2 describe the structural equation model, Section 2.3 explain the exploratory factor analysis, and Section 2.4 describe the confirmatory factor analysis, as the mathematical methods that will be used. Section 3 presents the materials and methods applied. Section 4 presents the instrument validation. Section 5 presents a brief discussion of the results and implications. Finally, the conclusions and recommendations are presented in Section 6.

## 2. Theoretical Background

### 2.1. Dimensions of Ergonomics and Hypotheses Development

Ergonomics has been universally used to improve the quality of human life, such as health, safety, comfort, and productivity so that the personnel is satisfied in the environments and work activities. In general, ergonomics can be analyzed considering two approaches (i.e., dimensions), the organizational dimension (organizational ergonomics, OE) and the physical dimension (physical ergonomics, PE), which are highly related to organizational sustainability. Organizational ergonomics, also called macroergonomics, refers to the optimization of social–technical systems, including their organizational structures, policies, and processes [16,17,18]. The relevant issues include communications, management of resources, labor projects, temporal work organization, teamwork, participative project, new work paradigms, cooperative work, organizational culture, network organizations, and quality management. The macroergonomics approach satisfies the correct criteria for the design of work systems and work designs with the man–system interface; focused on the man, applying a “humanized” task to the role of different assignments [19]. Therefore, job designs also include work modules, tasks, knowledge, and skill requirements, as well as factors such as an autonomy degree, identity, feedback, and opportunities for social interactions. The purpose of organizational ergonomics involves the optimization of the design of social–technical work systems and the study of the effect of organizational structures on human behavior and safety. This goal has been achieved through the systematic consideration of the relevant variables of sociotechnical systems in the ergonomic analysis, design, implementation, evaluation, and control of the process. These variables correspond to the technological subsystem, personal subsystem, and external environment [20]. To achieve the maximum objectives of ergonomics (i.e., optimize the well-being of people and the overall performance of the system) the interaction between the subsystems has to be functional, respecting the capacities and limitations of the human being and his/her culture [21]. Considering the above, managers need to recognize that job satisfaction is a feeling of relative pleasure or pain [1]. Furthermore, this factor arises from the perception of how an employee views his/her job, positively or negatively, or how he/she likes or dislikes the job as a result of the employee’s perception of the job. To summarize, satisfaction is considered one of the most important pillars in assessing the success of companies and it can help management understand the reactions of workers to their jobs. Hence, the hypothesis related to job satisfaction and organizational ergonomics is presented:
**Hypothesis** **1** **(*H1*).***The ergonomic factors inherent to the **process** have positive impacts on the job satisfaction of workers who provide operational services in the institution.*

Regarding physical ergonomics, it is considered of interest for the anthropometry, anatomy, and biomechanics of man, concerning physical, mental, and environmental efforts. Moreover, it is a multidisciplinary model that is interested in the adaptation of work for man and how these relate to the physical activities involved in the use of the musculoskeletal and cardiovascular systems [22,23]. This construct was established as the science that studies the dimensions of the human body, the knowledge, and techniques to carry out measurements, as well as their statistical treatments [24]. Thus, it seeks to provide anthropometric data that serve as the bases for sizing objects that adjust to the true characteristics of end users. In addition, considering the reality of various industrial sectors, most work tasks require the worker to maintain a fixed posture for long periods, and if a poorly designed position is added to this, then it does not correspond to the anthropometric characteristics of the end users; consequently, it can encourage the adoption of uncomfortable postures, undue efforts, causing discomfort, fatigue in certain muscle groups, and health effects on workers [25]. Moreover, productivity as well as quality decrease. Finally, the probability of errors and the number of work accidents would increase. This field perceives the handling of loads, repetitive movements, and inadequate postures, which cause musculoskeletal difficulties and pathological alterations in health [22]. This construct has a considerable weight in satisfaction. Therefore, the forms that job satisfaction take in objective conditions are related to occupational safety and hygiene, workloads, occupational health, etc., and in subjective conditions of the worker in the sense of how they experience it. The occupational risks of physical loads should be included in the measurement of job satisfaction since it has been shown that a dissatisfied worker is more likely to suffer accidents. Ergonomics at work becomes relevant when identifying a situation that causes a deterioration in the employee’s health. Therefore, the following hypothesis is presented:
**Hypothesis** **2** **(*H2*).***The ergonomic factors inherent to the **operator** have positive impacts on the job satisfaction of the workers who provide operational services in the institution.*

In this research, the third hypothesis relates the two constructs of organizational and physical ergonomics—to determine if there is a relationship in both latent variables. Thus, the third hypothesis is:
**Hypothesis** **3** **(*H3*).***The ergonomic factors inherent to the **process** (related to OE) have positive impacts on the ergonomic factors inherent to the **operator** (related to PE) who provides operational services at the institution.*

Figure 1 shows the relationship between the mentioned hypotheses.

### 2.2. Structural Equation Model

The structural equation model (SEM) is a multivariate technique that is applied in research in various disciplines due to its ability to explain causal relationships between qualitative and quantitative variables to test theoretical models [26]. The main contribution of SEM is that it allows researchers to evaluate theoretical models; it is becoming one of the most powerful tools for the study of causal relationships on non-experimental data when these relationships are linear [27]. This model designates a set of procedures and techniques for multivariate statistics that include a large number of classical methods, such as linear regression, exploratory and confirmatory factor analysis, and path analysis, among others [28]. However, its main feature involves the possibility of including unobservable variables in a much wider variety of models. Thus, it is a clear and objective tool for the empirical testing of theoretical hypotheses, being used in several disciplines, such as economics, psychology, sociology, education, marketing, etc. [29,30,31,32]. In this research, as part of SEM, we use (as a first step) the exploratory factor analysis and, consequently, the confirmatory factor analysis to eliminate those variables that do not have strong correlations in the study and determine which items of the ergonomic aspects have more relations and impacts with job satisfaction.

### 2.3. Exploratory Factor Analysis

To explore the latent variables more precisely, we use the exploratory factor analysis (EFA), which is one of the most frequently applied techniques in studies related to the development and validation of different areas since it explores the set of latent variables or common factors that explain the responses of the items, which are revealed from the observed variables [33,34]. In particular, the objective of this technique was to analyze and validate factors underlying a large set of data and reduce a large number of operational indicators into smaller numbers of conceptual variables [35]. Therefore, we chose—as a criterion—to accept those items whose values are greater than or equal to 0.5 [36]. Moreover, this multivariate method allows for group variables (e.g., items) that are strongly correlated with each other, and whose correlations with the variables of other groups (factors) are lower. Although the variables used are generally continuous, it is also possible to use this method on categorical variables [37]. According to [38], through the EFA, the score variability of a set of variables is explained by a smaller number of dimensions or factors. In this way, for example, a large number of items can be reduced to a small number of factors or dimensions that confer a theoretical meaning to the measurement. Each of these factors can group the intercorrelated items that are, at the same time, relatively independent from others sets (factors) of items. In addition, another factor that currently affects job satisfaction (in all sectors of society and industry) is related to the SARS-CoV-2 pandemic [39,40,41]. In particular, several workplaces have analyzed this aspect. However, none are based on the exploratory factor analysis.

### 2.4. Confirmatory Factor Analysis

The confirmatory factor analysis (CFA) allows correcting or corroborating if there is a deficiency of the EFA, leading to further testing of the specified hypotheses [42]. It also analyzes the covariance matrix instead of the correlation matrix, which helps to establish whether the indicators are equivalent [43]. In particular, The CFA is represented by flow diagrams (path diagrams), according to its particular specifications (see Figure 2). The rectangles represent the items and the ellipses, the common factors. The unidirectional arrows between common factors and items express saturations and the bidirectional arrows indicate the correlation between common or unique factors [44]. Thus, this method provides the statistical framework to evaluate the validity and reliability of each item instead of performing a global analysis, helping the researcher to optimize both the construction of a measurement instrument and the analysis of results [45]. The mathematical equation of the model is described by *n* = $\lambda_{i}\xi_{i}+\delta_{i}$, where *n* is the number of all the observable variables that appear in each dimension. In this study, 31 items are presented as observable variables, $\lambda_{i}$ is the factor load that explains the item concerning the latent variable, $\xi_{i}$ is the latent variable, in which this research has three dimensions (job satisfaction related to physical and organizational ergonomics), and finally $\delta_{i}$ is the error of the observable variable. Moreover, it is important to mention that no studies were found that relate ergonomic factors in the three dimensions mentioned in the structural equation model framework. Furthermore, there are no works related to job satisfaction research educational institutions using this model during the SARS-CoV-2 pandemic. Considering the above, particularly the lack of projects related to institutions in the educational sector, the purpose of this study was to identify which factors have strong correlations to determine the impact of job satisfaction (JS) concerning ergonomic factors (organizational and physical dimensions), using (as study objects) the unionized workers at the University of Sonora, South Campus, through a cross-sectional exploratory factor analysis.

## 3. Materials and Methods

For this work, the study design was a non-experimental quantitative type, cross-sectional, with a correlational scope; the study objects were unionized workers who provide operational manual services at a public higher education school. STATA 14 software (StataCorp LLC, College Station, TX, USA) was used for the descriptive data analysis and the implementation of different estimation techniques. The number of employees integrated is represented by a population of 92 unionized people. Furthermore, the sample size was determined to know the exact number of participants to be included in job satisfaction research, which was divided into groups with different degrees—from the basic level to the undergraduate level using the finite population formula, as Table 1 shows. Thus, stratified sampling was selected because the study object had the same chance of being selected randomly. Consequently, the standard deviation (*sh*) was used in the stratified sample to know the exact number of workers for each category, where *n* is the sample of the study (75 unionized people); it was divided into *N* (the population of 92 unionized people) to have 0.815 (*n*/*N*). As a result, the number of people by category who answered the questionnaire designed in a structured interview framework for this research is shown in Table 2.

Once the stratified samples were determined, we then designed our customized measuring instrument to determine job satisfaction and occupational health, which consisted of closed-type questions, with nominal coding levels for demographic variables, ordinal coding for the constructs of job satisfaction, and the ergonomic aspects inherent to the operator and organizational process, with a Likert scale of five response levels (from descending to ascending scales): totally dissatisfied (1), dissatisfied (2), neither satisfied nor dissatisfied (3), satisfied (4), and totally satisfied (5). Next, a pilot test on the sample was performed to validate the research instrument using a small portion of 10 randomly selected people representing 13% of the sample with equivalent characteristics, applying Cronbach’s alpha coefficient method. Furthermore, it allowed measuring the confidence level, mostly with a quantitative approach, with a confidence level of 95% [46]. Table 3 shows Cronbach’s alpha magnitude values. This type of analysis is a statistical tool that supports the researcher in order to know the magnitude of the reliability of the study based on the average of the correlations between the indicators. If the coefficient is low (0.21–0.40), the constructs should be reconsidered and improved.

As a result, it was determined that the instrument (conformed by 31 items) was highly reliable with a significance value of 0.05, considering an overall Cronbach´s alpha coefficient of 0.9028, i.e., a very high magnitude (in certain contexts and by tacit agreement, it was considered that alpha values greater than 0.7–0.8 were adequate to guarantee the reliability of the scale). In particular, the overall Cronbach´s alpha coefficient was the average value of the particular Cronbach´s alpha coefficient for each dimension (i.e., organizational ergonomic, physical ergonomic, and job satisfaction), 0.935, 0.8653, and 0.9031, respectively. Hence, we obtained a clear summary of the data, described the key trends in the study objects, and observed the situations that led to new facts (that were part of the investigation). Thus, the results obtained in the survey are presented below. Once the data provided through the surveys were examined, it can be seen that of the 75 workers (sample), there were 24 female workers (32%) and 51 male workers (68%) (see Table 4). Moreover, Table 5 shows the cumulative frequency considering the type of contract. Thus, 88% of the workers had indefinite contracts (i.e., bases), representing 66 people; moreover, 12% were temporary (only 9 from a total of 75 workers).

Regarding schooling, it is one of the most important requirements for hiring in some important positions within the institution. It can be seen in Table 6 that, 58.67% (cumulative frequency) of the workers have basic level studies (i.e., primary and secondary academic level). Moreover, 20% have obtained upper secondary level studies and 21% have higher-level studies. Moreover, secondary education represents the highest percentage (i.e., 43%).

For this research work, three latent variables were considered—job satisfaction and physical and organizational ergonomic aspects. As part of the analysis, the determination of strong correlation variables (factorial load values higher than 0.50) was performed. As aforementioned, we examined (and had greater precision) of those indicators that could serve as reliable data for decision-making about job satisfaction [35]. Consequently, Table 7, Table 8 and Table 9 show the 31 indicators that made up the survey, of which 10 items correspond to job satisfaction (Q1_JS–Q10_JS) [47], 8 items relate to physical ergonomics (Q11_PE–Q18_PE), and 13 items concern organizational ergonomic constructs (Q19_OE–Q31_OE).

The survey also included some demographic variables (gender, antiquity, schooling level, place of birth, functional area, age, type of contract, civil status, place of residence, and salary), since they are factors that conform to the different characteristics of human populations [48].

## 4. Instrument Validation

The correlation calculation supported the validity verification of the study concerning the construct of the job satisfaction (*JS*) variable to the ergonomic aspects inherent to the operator (physical) and the processes (organizational). Table 10 shows the Pearson correlation and significance level results concerning the construct and the dimensions mentioned. The significance level (*p*) was typically set no higher than 0.05.

Regarding the job satisfaction–organizational ergonomics relationship, it presented a correlation coefficient (Pearson correlation) value of 0.707 (high and significant positive correlation), which means that the more this ergonomic aspect is linked, the more the workers will be satisfied in this institution. Consequently, it has high validity in the relation between these two variables. Moreover, the Pearson correlation between job satisfaction and physical ergonomics is 0.628, which means that the more the conditions for physical workloads are improved, the better the worker´s job satisfaction will be. Likewise, these two variables had a high validation in the investigation. Moreover, the relationship between organizational ergonomics and physical ergonomics showed a moderate positive correlation of 0.552 with a significant level and it was directly proportional to the two variables. Hence, if the perceptions of both variables increase, employee satisfaction will increase.

## 5. Results and Discussion

Once the results related to the descriptive analyses of the variables were obtained, the next step was to determine the items that belonged to each construct through the component matrix, in order to establish the appropriate instrument. For this, the criterion was taken to accept those items whose values were greater or equal to 0.5, since explanatory capacity was gained.

Table 11 shows the results, taking into account the KMO–Bartlett test, where 13 observable variables were eliminated from the 31 original items, i.e., 31 items were analyzed for EFA. Therefore, the overall KMO–Bartlett test of the instrument was 0.789, with the KMO–Bartlett value of the JS, PE, and OE being 0.744, 0.798, and 0.825, respectively. To clarify, regarding the job satisfaction (JS) variable, out of a total of 10 items, only 6 of these had strong correlation coefficients between Q6_JS, Q3_JS, Q1_JS, Q4_JS, Q7_JS, and Q5_JS. While the items Q2_JS, Q8_JS, Q9_JS y Q10_JS were eliminated because these variables obtained values less than 0.5. Concerning the physical ergonomic (PE) construct, out of about eight items, only five (Q17_PE, P18_PE, Q15_PE, Q14_PE, and Q13_PE) had strong correlation coefficients between the reagents. While the items Q11_PE, Q12_PE y Q16_PE were eliminated using the same elimination rule mentioned. Similarly, for the organizational ergonomic (OE) dimension, seven items (Q24_OE, Q26_OE, Q28_OE, Q19_OE, Q23_OE, Q27_OE, and Q25_OE) had strong correlations, deleting the items, Q19_OE, Q20_OE, Q21_OE, Q22_OE, Q29_OE, and Q30_OE using the same criteria. In the previous procedure, the exploratory factor analysis was developed to know and adjust the items that support the construction of the proposed model, resulting in a total of 18 endogenous variables among the three constructs. Once this part of the study was completed, the next step was to apply the confirmatory factor analysis using the 18 items that were the EFA results. This analysis made it possible to corroborate or correct (if needed) the deficiency of the FEA, leading to further testing of the specified hypotheses [42]. In the CFA, it is necessary to observe the factor loadings that allow the correlation between variables and factors to be established. The closer they are to the unit (1), the higher the correlation. A rule of thumb in the CFA states that loadings must be ≥0.3 in the absolute value considered optimal [36,37]. Consequently, Figure 2 shows the first test of the proposed model as a whole of the constructs with the observable variables, of which the physical ergonomics variable had five items; concerning organizational ergonomics—seven items; finally, job satisfaction was represented by six items, for the verification of the hypotheses raised. The following model was developed, taking into account the results of the present research.

For the model validation, the observable variables having loading ≥0.3 with the CFA were considered. Table 12 presents the criteria for each absolute fit index to verify whether the study has a good or acceptable fit to the model, which means a *p*-value of less than 0.05.

A detailed analysis of the proposed model was carried out in order to verify which observable variables were relevant in each construct for the validation of the hypotheses put forward. Hence, from a total of 18 variables (considering 31 items as inputs) resulting from the exploratory factor analysis, 6 of them were eliminated in the first test of the confirmatory factor analysis for having low factor loadings of 0.3 and also for not complying with adequate adjustments considering the diverse goodness of fit analysis. These were Q1_JS, Q3_JS, Q5_JS, Q14_EF, Q23_EO, and Q28_EO. In the second test of the confirmatory factor analysis, 12 variables as inputs were considered, and then the goodness of fit and the hypotheses were analyzed to determine whether the model was accepted or rejected. Thus, Figure 3 shows the final test (based on CFA) of the proposed model. To support the study, chi-square was used as a hypothesis test, which compared the observed distribution with an expected distribution of the data, whose purpose was to test the relationship between the two variables. The goodness of fit index (GFI) is an index used to measure and compare the discrepancies between various constructs in a fitted model, and standardized (SRMR) refers to the standardized root mean square residual, which indicates that if the index is closer to 0, the model will be better.

### 5.1. Goodness of Fit Results

With respect to the results obtained in the second test of the model, the evaluation of the goodness of fit is that the RMSEA has a value of 0.036, and a chi-square (*x*^2^) of 1.09 degrees of freedom (*df*), which means that both have a good fit in the model. In addition, the standardized root mean square (SRMR) presents an index of 0.066, and the goodness of fit index (GFI) of about 0.900, which infers that the study has an acceptable fit in the research. Consequently, it presents favorable results concerning the variables proposed in the model. Moreover, the results of the hypotheses show good results in the hypotheses proposed, as Table 13 shows for *p* < 0.05.

### 5.2. Hypothesis Analysis

In the construct, organizational ergonomics were correlated with five indicators, which were: communication with managers (Q19_OE), new paradigms at work (Q24_OE), suggestions and ideas taken in the work team (Q25_OE), quality improvement initiatives (Q26_OE), and the way the institution was managed (Q27_OE), with regression weights values (λ) of 0.71, 0.58, 0.68, 0.80 and 0.46, respectively. In this dimension, the variables that most explain the organizational ergonomics were related to the items: quality improvement initiatives (related to Q27_OE) and communication with managers (related to Q19_OE). Therefore, to achieve quality improvement, the institution must implement good communication between the employee and the bosses because otherwise this variable would be affected; consequently, there would be a lack of quality improvement. On the other hand, the suggestions and ideas item was used by the work team (related to Q25_OE), in this aspect, the institution should take into account the ideas provided by the workers since they are generally aimed at improving the quality of work within the institution. Finally, there is a significant effect between EO and JS, resulting in a *p*-value of 0.000. As a result, the EO has a direct positive relationship and impact with JS because the standardized regression coefficient is *β* = 0.53. The aforementioned indicates that Hypothesis 1 (*H1*) is accepted.

According to the physical ergonomics latent variable, it is correlated with four indicators, which are: repetitive movements in the workplace (Q13_PE), physical stress, such as strain, neck, shoulder, and back loads (Q15_PE), workloads are well distributed (Q17_PE), and physical demands in the workplace (Q18_PE), where the results in regression weights (λ) are 0.50, 0.63, 0.79 and 0.80, respectively. Considering all the variables in this construct, the items that most explain physical ergonomics are the physical demands in the workplace (related to Q18_PE), and the distribution of the workload (related to Q17_PE). Thus, with respect to the physical demands that are contemplated in the institution, they must be assessed for each employee in the corresponding work area to avoid injuries that could put both the worker and the institution at risk. On the other hand, the institution must take care of the distribution of workloads since they must be equitable for each worker. Therefore, there is a significant effect between PE and JS, given that the *p*-value is 0.016. This means that physical ergonomics has a direct positive relationship with job satisfaction because the standardized regression coefficient is 0.37 (i.e., *β* = 0.37). The aforementioned indicates that Hypothesis 2 (*H2*) is accepted.

Finally, the relationship between the latent variables—organizational ergonomics and physical ergonomics—is significant (i.e., 0.000). Therefore, OE has a direct positive relationship with PE (i.e., *β* = 0.53). In addition, it can be mentioned that OE has a positive indirect relationship with JS through PE. Thus, the aforementioned indicates that Hypothesis 3 (*H3*) is accepted.

### 5.3. Impact on the Organizational Sustainability and Implications

This study has important implications for organizational sustainability policy and practice. In fact, it allows establishing a quantitative tool that helps analyze job satisfaction as a basic element for the development of particular guidelines in the financial, social, and environmental spheres. The foregoing implies that any initiative related to sustainable development that does not have the appropriate impetus on the part of the staff would be destined to fail. However, as many international standards exemplify (e.g., ISO-50001, energy management system), the encouragement and participation of staff are crucial for the success of sustainable development in a leadership framework. In addition, this study makes it possible to significantly impact occupational safety, and it addresses sustainable development from a more human perspective.

Regarding the theoretical and methodological implications, this study contributes to the organizational sustainability literature via a formal mathematical analysis of a particular business sector, in our case, the educational sector. However, this contribution can easily be adapted to other business sectors since the ergonomic dimensions analyzed are inherent to people and organizations. In particular, according to the particular necessities, some improvements in the method and instruments are feasible. Regarding the methodological implications, it is important to mention that within the organizational sustainability framework, the methods that support this development must be continuously enhanced. In this way, this study proposes a formal mathematical analysis that could help the permanent monitoring of the latent variables mentioned as part of the organizational sustainability methodology adopted by each company.

In the same sense, this study also presents very interesting practical implications. For example, the constant monitoring of the mentioned latent variables would allow knowing the job satisfaction level of particular periods and, thus, establish a job satisfaction baseline, which can serve to establish actions as countermeasures to improve job satisfaction level.

Finally, this study presents some limitations that establish potential future research. One of the main aspects is the number of people considered in the study. This implies that it is not possible to increase the organizational sustainability level (considering a baseline) only by analyzing a particular group of personnel; that is, the study must be extended to more workers and from other departments.

### 5.4. Impact on the Occupational Health

The above results allow us to infer that the work environment during the pandemic affected the job satisfaction of the interviewees. In particular, changes made during the pandemic related to physical and organizational ergonomics that affected job satisfaction. Although a specific study related to cognitive ergonomics was not carried out, which studies the cognitive aspects of workers and the interactions with the work system, qualitatively, it was perceived that job stress levels increased during the pandemic, which is related to the variables that most impact job satisfaction. In the same sense, the educational institution did not carry out a formal deployment of coping strategies against stress in workers, which was reflected in the results of job satisfaction. In addition, the results shown in Table 11 regarding physical ergonomics imply that the constructs (i.e., Q11_PE: Safety in your workplace, Q12_PE: Hygiene in your workplace, Q16_PE: Workplace design and Q18_PE: In general, how satisfied are you with the physical demands in your workplace?) do not have greater relevance for occupational health and job satisfaction. The foregoing imposes important organizational challenges regarding risk analysis. In a similar sense, regarding organizational ergonomics, the constructs that are least perceived as important for job satisfaction and occupational health are L Q20_OE: The communication with colleagues, Q21_OE: Teamwork, Q22_OE: Your work Schedule, Q29_OE: In my job, I can develop my skills, Q30_OE: I receive information on how I perform my work, and Q31_OE: Medical Services. Being the Q31_OE is more surprising.

## 6. Conclusions

The structural equation model is a useful tool that allows us to identify and group indicators that are strongly correlated with each other to reduce variables that do not contribute significantly to the study. With the development of this research, it was possible to know the parameters that have close relationships with the dimensions already raised during the SARS-CoV-2 pandemic. Nowadays, there is a lack of studies about the dimensions of ergonomics regarding job satisfaction–occupational health and other variables. Thus, our results open the door to developing other multivariate statistical methods as the next step to have a more in-depth analysis of the constructs of job satisfaction concerning other ergonomic aspects, such as cognitive and temporary ergonomics. Moreover, the results presented can be applied as part of the design, planning, and management of technical and social systems at any organization. Thus, this study can be applied to the managers of institutions in order for them to know the perceptions of their workers who provide operational services. Therefore, it would be a matter of interest to know the perceptions or feelings of both parties. Finally, this methodology and finding can be considered as a background to other sectors, such as manufacturing, agroindustry, and health, among others. One important aspect to clarify is that the sample size used in this project may appear to be small (i.e., this is a potential limitation). However, the sample size is relative to the model complexity, as well as to the a priori existence of the strong theory related to the instrument to be validated [49,50]. In other words, in our case, there is no solid a priori theory or a similar instrument (which is a limitation); for this reason, the sample was small. In addition, the said sample can be used for both the EFA and CFA, without losing reliability as part of the validation of our instrument. As part of future work, this proposed instrument should be further analyzed with other samples using only the CFA. Regarding the theoretical implications, our work presents evidence of an instrument with multiple ergonomic dimensions, which was not found in the literature concerning job satisfaction and occupational health. Moreover, regarding the practical implications, the results of this work can be used to promote and maintain higher degrees of physical, mental, and social well-being for workers as part of occupational health and job satisfaction.

## Figures and Tables

**Figure 1 ijerph-19-10714-f001:**
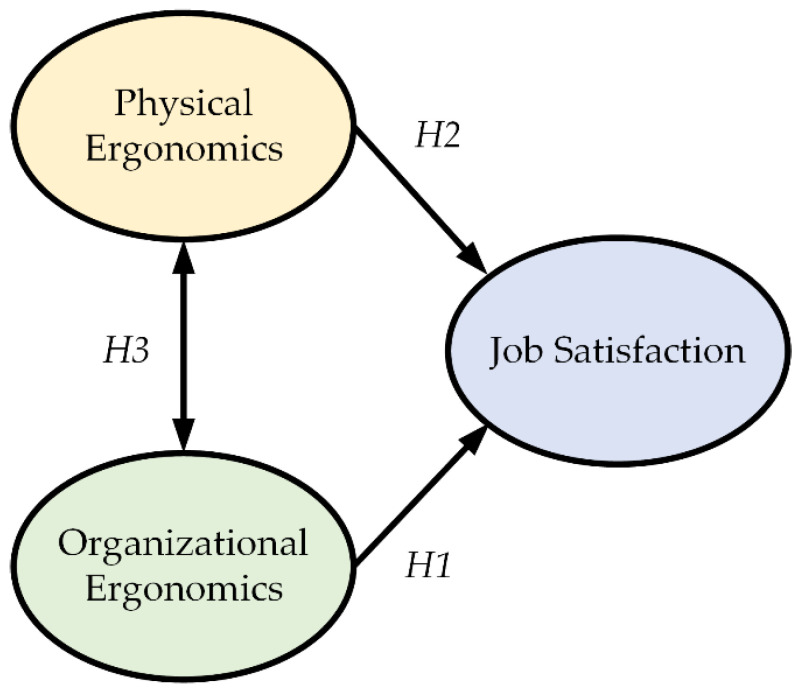
Relationship between ergonomic aspects and job satisfaction.

**Figure 2 ijerph-19-10714-f002:**
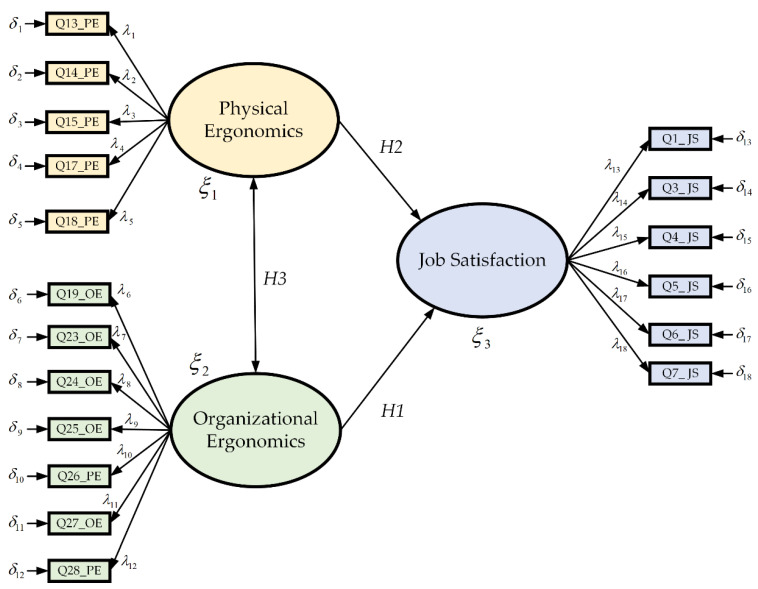
The first test of the model.

**Figure 3 ijerph-19-10714-f003:**
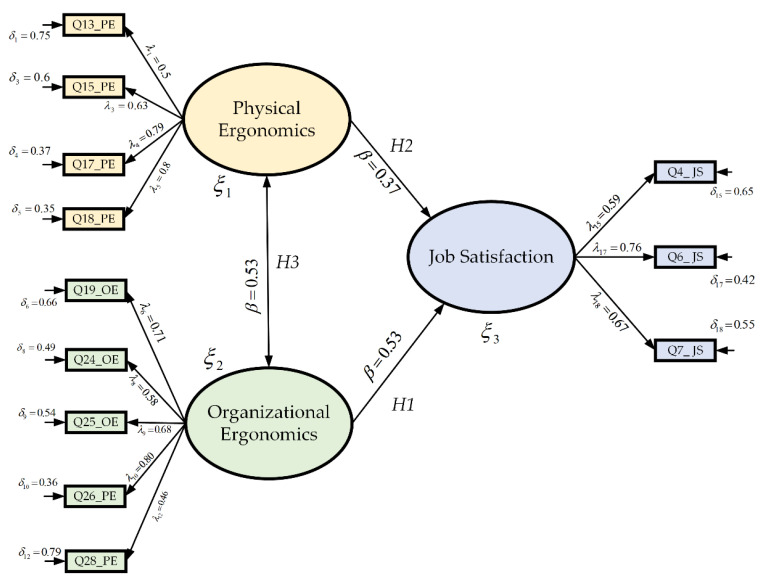
The final test concerning JS with PE and OE.

**Table 1 ijerph-19-10714-t001:** Subpopulation samples for educational levels based on the Mexican education system.

Educational Level	Amount of People (*Nh*)
Primary (MX)/Elementary (US)	14
Secondary (MX)/Middle (US)	40
Preparatory (MX)/High School (US)	19
Undergraduate (MX)/College (US)	19
**Total**	**92**

**Table 2 ijerph-19-10714-t002:** Stratified probability sampling.

Stratum Number	Schooling Level	Subgroup Total	Sample =(sh)(Nh)	Sample
1	Primary (MX)/Elementary (US)	14	(0.815) (14)	11
2	Secondary (MX)/Middle (US)	40	(0.815) (40)	33
3	Preparatory (MX)/High School (US)	19	(0.815) (19)	16
4	Undergraduate (MX)/College (US)	19	(0.815) (19)	16
**Total**		**92**		**75**

**Table 3 ijerph-19-10714-t003:** The Cronbach alpha magnitude values.

Rank	Magnitude
0.81–1.00	Very high
0.61–0.80	High
0.41–0.60	Moderate
0.21–0.40	Low
0.001–0.20	Very low

**Table 4 ijerph-19-10714-t004:** Cumulative gender frequency.

Gender	Frequency	%	Total
Women	24	32	32%
Men	51	68	100%
**Total**	75	100	

**Table 5 ijerph-19-10714-t005:** Cumulative frequency of the contract type.

Contract	Frequency	%	Total
Base	66	88	88%
Eventual	9	12	100%
**Total**	75	100	

**Table 6 ijerph-19-10714-t006:** Cumulative frequency of schooling.

Schooling Level	Frequency	%	Total
Primary (MX)/Elementary (US)	12	16	16%
Secondary (MX)/Middle (US)	32	43	58.67%
Preparatory (MX)/High School (US)	15	20	78.67%
Undergraduate (MX)/College (US)	16	21	100%
**Total**	75	100	

**Table 7 ijerph-19-10714-t007:** Constructs of job satisfaction.

Dimension	Items
**Job Satisfaction**	Q1_JS: Relationship between boss and worker. Q2_JS: Relationship between workers. Q3_JS: The supervision and guidance of the boss. Q4_JS: Recognition at work. Q5_JS: Freedom in choosing the working method. Q6_JS: Perceived work environment. Q7_JS: Labor condition at work. Q8_JS: The position you hold, you consider. Q9_JS: Salary. Q10_JS: Benefits.

**Table 8 ijerph-19-10714-t008:** Constructs of physical ergonomics.

Dimension	Items
**Physical Ergonomic**	Q11_PE: Safety in your workplace. Q12_PE: Hygiene in your workplace. Q13_PE: Repetitive movements in your workplace. Q14_PE: Carry or move objects. Q15_PE: Physical loads in terms of strength, neck, shoulder, and back. Q16_PE: Workplace design. Q17_PE: Workloads are well distributed. Q18_PE: In general, how satisfied are you with the physical demands in your workplace?

**Table 9 ijerph-19-10714-t009:** Constructs of organizational ergonomics.

Dimension	Items
**Organizational Ergonomic**	Q19_OE: The communication with bosses. Q20_OE: The communication with colleagues. Q21_OE: Teamwork. Q22_OE: Your work Schedule. Q23_OE: My roles and responsibilities are well defined. Q24_OE: The new paradigms in my work. Q25_OE: Suggestions and ideas are taken into account in my work team. Q26_OE: Quality improvement initiatives. Q27_OE: The way in which the institution is managed. Q28_OE: Resource management. Q29_OE: In my job, I can develop my skills. Q30_OE: I receive information on how I perform my work.Q31_OE: Medical Services.

**Table 10 ijerph-19-10714-t010:** Pearson correlation coefficient calculation concerning the constructs based on the sample (75 workers).

Variable	OE	JS	PE
Organizational ergonomic (*OE*)	1	0.707 *	0.552 *
Job satisfaction (*JS*)	0.707 *	1	0.628 *
Physical ergonomic (*PE*)	0.552 *	0.628 *	1

* *p* < 0.05.

**Table 11 ijerph-19-10714-t011:** Component matrix per construct.

Items	JS	PE	OE
Q6_JS	0.807		
Q3_JS	0.783		
Q1_JS	0.765		
Q4_JS	0.731		
Q7_JS	0.643		
Q5_JS	0.565		
Q17_PE		0.809	
Q18_PE		0.791	
Q15_PE		0.741	
Q14_PE		0.699	
Q13_PE		0.691	
Q24_OE			0.750
Q26_OE			0.747
Q28_OE			0.729
Q19_OE			0.689
Q23_OE			0.671
Q27_OE			0.669
Q25_OE			0.667

**Table 12 ijerph-19-10714-t012:** Absolute fit index.

Absolute Fit Index	Good Fit	Acceptable Fit
$\mathrm{Chi}-\mathrm{square}\;(x^{2})/df$	0 ≤ *x*^2^ ≤ 2*df*	2*df* ≤ *x*^2^ ≤ 3*df*
Root mean square error of approximation (*RMSEA*)	0 ≤ *RMSEA* ≤ 0.05	0.05 ≤ *RMSEA* ≤ 0.08
Goodness of fit index (*GFI*)	0.95 ≤ *GFI* ≤ 1.00	0.90 ≤ *GFI* ≤ 0.95
Standardized (*SRMR*)	0 ≤ *SRMR* ≤ 0.05	0.05 ≤ *SRMR* ≤ 0.10

**Table 13 ijerph-19-10714-t013:** Model regression coefficients.

Model Regression Coefficients	*β*	*p*
Job Satisfaction ← Organizational Ergonomic	0.53	0.000
Job Satisfaction ← Physical Ergonomic	0.37	0.016
Physical Ergonomic ← Organizational Ergonomic	0.53	0.000

## Data Availability

Not applicable.

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
