# Peer review of "Ergonomic Factors That Impact Job Satisfaction and Occupational Health during the SARS-CoV-2 Pandemic Based on a Structural Equation Model: A Cross-Sectional Exploratory Analysis of University Workers"

_ijerph, 2022, doi:10.3390/ijerph191710714_

Round 1
Reviewer 1 Report
The paper is written with logical sequence and good flow. Authors have demonstrated a good understanding of the research area and methods of data collection. I was very impressed with the sections on discussion on findings and their significance.
However, I would like the authors to make a few improvements on the current draft:
1) The research problem needs to be highlighted in the introduction section. I read it a few times and yet failed to understand the reserach problem that was investigated. Some kind of data/evidence of the research premise blending with literature gap would augur very well here.
2) Authors used Cronbach Alpha to test the internal consistency of the data. However, authors used all items together to run statistical tests to get the Cronbach Alpha which is not right. There are three variables and authors must run them separately to test the internal consistency of items. Therefore, there would be three Cronbach Alpha.
3) Authors developed structural equation modelling without mentioning which software was used. If they have used SmartPLS, I would then expect them to run Measurement model followed by the Structural model. I could not find that flow in data analysis.
With these minor corrections, I recommend this paper to be accepted for publication into IJERPH.
Author Response
REVIEWER 1
REVIEWER’S COMMENTS #1: The research problem needs to be highlighted in the introduction section. I read it a few times and yet failed to understand the research problem that was investigated. Some kind of data/evidence of the research premise blending with literature gap would augur very well here
RESPONSE TO REVIEWER #1: We respectfully believe that the need for the study is made clear in the Introduction section, specifically in Lines 96-102, “…The literature shows that the study of job satisfaction has been approached from multiple dimensions, which is linked to some variables such as motivation, sociocultural aspects, and economic and communication features, among others [10]. But, it has not been addressed with ergonomic approaches in higher-level institutions focused on staff who are unionized people who provide operational service (maintenance, security, hygiene in the infrastructure, drives, and others) because this scenario is not widely studied, i.e., case studies and related information are scarce…”
In fact, the Theoretical Background section also supports the need mentioned, Lines 248-250, “…Furthermore, there are no works related to job satisfaction research educational institutions using this model during the SARS-CoV-2 pandemic. Considering the above, particularly the lack of projects related to institutions in the educational sector…”
Respect to the comment, “The research problem needs to be highlighted in the introduction section.” After a review of the document, it was determined that the objective/ research problem of the document is already present in lines 105-107, “…Therefore, this study aims to determine the impact of ergonomic factors on job satisfaction and occupational health concerning unionized workers who provide operational services at this institution through structural equation analysis.”
REVIEWER’S COMMENTS #2: Authors used Cronbach Alpha to test the internal consistency of the data. However, authors used all items together to run statistical tests to get the Cronbach Alpha which is not right. There are three variables and authors must run them separately to test the internal consistency of items. Therefore, there would be three Cronbach Alpha.
RESPONSE TO REVIEWER #2: The reviewer's comment is highly pertinent. We only show the overall Cronbach's Alpha coefficient, but we do not mention the particular Cronbach's Alpha values per dimension. Considering the above, we made some modifications to the document. Please check the following lines:
Line 303-310, “…As a result, it is determined that the instrument (conformed by 31 items) is highly reliable with a significance value of 0.05 considering an overall Cronbach´s Alpha coefficient of 0.9028, i.e., a very high magnitude (in certain contexts and by tacit agreement, it is considered that alpha values greater than 0.7-0.8 are adequate to guarantee the reliability of the scale). In particular, the overall Cronbach´s Alpha coefficient is the average value of the particular Cronbach´s Alpha coefficient for each dimension (i.e., Organizational Ergonomic, Physical Ergonomic, and Job Satisfaction), 0.935, 0.8653, and 0.9031, respectively.”
Also, the abstract section was modified.
REVIEWER’S COMMENTS #3: Authors developed structural equation modelling without mentioning which software was used. If they have used SmartPLS, I would then expect them to run Measurement model followed by the Structural model. I could not find that flow in data analysis.
RESPONSE TO REVIEWER #3: The comment is very valid to clarify the research design to the reader. In this way, the STATA 14 software was used in our manuscript. We add the following information: Lines 258-260, “…In particular, the STATA 14 software was used for descriptive data analysis and the implementation of different estimation techniques…”

Reviewer 2 Report
Dear Author,
Please find attached my review report.
Wish you the best of luck in future!

Author Response
REVIEWER 2
REVIEWER’S COMMENTS #1: Title: It mentions the SAR-CoV 2 but fails to make the case throughout the article.
RESPONSE TO REVIEWER #1: We respectfully consider that our article does present the case. Next, we show some lines of the manuscript that are evidence about our argument.
Lines 204-206 mention, "...In addition, another factor that has currently affected job satisfaction in all sectors of society and industry is related to the SARS-CoV-2 pandemic [39-41]". Also, Lines 248-254 mention, "Furthermore, there are no works related to job satisfaction research educational institutions using this model during the SARS-CoV-2 pandemic. Considering the above, particularly the lack of projects related to institutions in the educational sector, the purpose of this study is to identify which factors have a strong correlation to determine the impact of Job Satisfaction (JS) concerning ergonomic factors (organizational and physical dimensions) having as study object the unionized workers of the University of Sonora, South Campus, through the Cross-Sectional Exploratory Factor Analysis."
REVIEWER’S COMMENTS #2: We consider that the manuscript produced confusion in the reviewer. Next, we will try to clarify the doubt of the reviewer. Abstract: In the abstract, the workers of the University of Sonora have been mentioned as the respondents, while in the Materials and Methods, the respondents are school employees with education levels of primary, secondary, etc., have been mentioned. Moreover, the abstract has not been written in a standard way; goodness of fit, reliability values, etc., are not mentioned.
RESPONSE TO REVIEWER #2: In the Abstract, Lines 253-254 mentions, "...using the unionized workers of the University of Sonora as the object of study...", which is taken up in the Materials and Methods section, Lines (236-237). Later, in the Materials and Methods section, this population was segmented by academic level, see Tables 1 and 2. Therefore, the respondents are unionized workers, who were segmented by academic level. Our manuscript never mentions, "...the respondents are school employees with education levels of primary, secondary, etc., have been mentioned..."
In addition, the structure and content of the abstract were improved. Please check out the following particular sentences in the abstract section:
“…As result, overall Cronbach's Alpha coefficient was determined, 0.9028, which is considered adequate to guarantee reliability (i.e., very highly magnitude). Therefore, after the Structural Equation Model, only 12 items present a strong correlation which has a good model fit of 0.036 based on the Root Mean Square Error of Approximation, 1.09 degrees of freedom for the Chi-square and 0.9 for Goodness of Fit Index, with a confidence level of 95 %...”
“…Where organizational, and physical factors have a positive impact on job satisfaction with a factor load of 0.37 and 0.53, respectively, with a p-value of 0.016 and 0.000, respectively.”
“…In fact, there exists a knowledge gap about the relation between diverse ergonomics factors, job satisfaction, and occupational health, in the educational institution's context…”
REVIEWER’S COMMENTS #3: Introduction: It fails to create the need for the study by discussing the research gap.
RESPONSE TO REVIEWER #3: We respectfully believe that the need for the study is made clear in the Introduction section, specifically in Lines 96-102, “…The literature shows that the study of job satisfaction has been approached from multiple dimensions, which is linked to some variables such as motivation, sociocultural aspects, and economic and communication features, among others [10]. But, it has not been addressed with ergonomic approaches in higher-level institutions focused on staff who are unionized people who provide operational service (maintenance, security, hygiene in the infrastructure, drives, and others) because this scenario is not widely studied, i.e., case studies and related information are scarce…”
In fact, the Theoretical Background section also supports the need mentioned, Lines 248-254, “…Furthermore, there are no works related to job satisfaction research educational institutions using this model during the SARS-CoV-2 pandemic. Considering the above, particularly the lack of projects related to institutions in the educational sector…”
REVIEWER’S COMMENTS #4: Theoretical Background: The heading should be extended by incorporating ‘Hypotheses Development’. Hypothesis H3 has not been developed at all.
RESPONSE TO REVIEWER #4: The reviewer is completely right. We extended the heading by incorporating "Hypotheses Development'. Now, the heading is, '2.1. Dimensions of Ergonomics and Hypotheses Development'.
Also, Hypothesis 3 is not explained in detail since the related constructors have already been mentioned in the hypotheses (H1 and H2) and ergonomic dimensions in relation to Job Satisfaction. In this way, the first two hypotheses (H1 and H2) are the most important in the study. Therefore, a third hypothesis (H3) is determined in order to know the relationship between these ergonomic dimensions (physical and organizational ergonomics).
REVIEWER’S COMMENTS #5: Exploratory Factor Analysis: Since the purpose of the study is not to develop the scale, this section is redundant. The EFA and CFA sections are general stuff (known to everyone).
RESPONSE TO REVIEWER #5: In general, the main uses of the EFA are 1) To guide the development of scales, 2) To analyze the behavior of each item, and 3) Reduce the size of the data (go from 10 items to 1 total score). In our case, the EFA was used for option 3. However, the original version of the manuscript did not present a final score as the output of the EFA. Please check: Lines 378-380, “…Therefore, the overall KMO-Bartlett's test of the instrument is 0.789, with the KMO-Bartlett value of the JS, PE, and OE being 0.744, 0.798, and 0.825, respectively.”
Regarding the EFA and CFA sections, we consider it pertinent to provide general information for the benefit of the future readers. Since this type of analysis has not been reported in the particular context of the article.
REVIEWER’S COMMENTS #6: Materials and Methods: The sampling procedure is missing; which schools were engaged, and how? The sample of 72 is insufficient to run and get reliable results employing SEM. Line 247: Interviews were conducted?? The reliability of each construct should be determined separately. The same data for EFA and CFA does not make sense. Out of 31, only 12 items were used for the analysis; the data collected may not be correct.
RESPONSE TO REVIEWER #6:
The reviewer comments, “The sampling procedure is missing; which schools were engaged, and how?...” The manuscript makes it clear in various sections and lines that the study was conducted for a single educational institution, for example, in the abstract section and Lines 249-250. We believe that the phrase in the title, “A Cross-Sectional Exploratory Analysis of Universities Workers”, may have been the phrase that caused the confusion. Regarding the sampling procedure, the Material and Methods section explains the procedure used. In particular, Lines 256-266, and Tables 1 and 2 provide details on sampling.
Regarding the comment, “…The sample of 72 is insufficient to run and get reliable results employing SEM…” Technically, the sample size used in SEM really depends on model complexity but also on many other factors (e.g., normality of the data, missing patterns, among others). That is, complex systems require larger samples than simple systems. Considering the above, regarding the sample size, various practical opinions and arguments exist in the literature. For example, Boomsma (1982), in “The robustness of LISREL against small sample sizes in factor analysis models” mentions that a minimum sample size of 100 or 200 is a good consideration. On the other hand, there are researches where sample sizes of 50-70 have been used, which have been sufficient to describe particular models. Please see the following reference: (https://doi.org/10.1177/0013164414525397)(https://www.tandfonline.com/doi/pdf/10.1207/s15327752jpa8501_02) Considering the above, we consider that a sample size of 75 is sufficient, in addition to considering that the population size is not large enough. Thus, in the conclusions section, arguments are provided to clarify the reviewer's concern. Please check out the Lines 559-572, “…An important aspect to clarify is that the sample size used in this project may appear to be small (i.e., this is a potential limitation). However, the sample size is relative to the model complexity, as well as to the a priori existence of strong theory related to the instrument to be validated [49-50]. In other words, in our case, there is no solid a priori theory or a similar instrument (which is a limitation), for this reason, the sample is small. In addition, said sample can be used for both the EFA and CFA, without losing reliability as part of the validation of our instrument. As future work, this proposed instrument should be further analyzed with other samples using only the CFA. Regarding the theoretical implications, our work presents evidence of an instrument with multiple ergonomic dimensions, which is not found in the literature concerning job satisfaction and occupational health. Also, regarding the practical implications, the results of this work can be used to promote and maintain a higher degree of physical, mental, and social well-being of workers as part of occupational health and job satisfaction.”
The reviewer comments, “…Line 247: Interviews were conducted?? The reliability of each construct should be determined separately…” After reviewing the writing, we consider that the idea was not clearly written. Therefore, we improve the structure and wording of Lines 270-272 as follows: ”… As a result, the number of people by category who answered the questionnaire designed in a structured interview framework for this research is shown in Table 2.” Regarding the reliability of the constructs, we consider that it is not pertinent to determine the reliability separately, since the aforementioned hypotheses and the analyzes shown are based on the set of constructs.
The reviewer comments, “The same data for EFA and CFA does not make sense. Out of 31, only 12 items were used for the analysis; the data collected may not be correct.”
The reviewer is correct that EFA and CFA generally do not use the same data. However, the above argument is recommended but not entirely needed. In particular, performing EFA and CFA in the same data set implies that no a priori solid theory about the instrument's structure that it intends to validate. This is our case. As we mentioned in the manuscript, after the literature review, no solid theory or similar instruments were found that analyze job satisfaction and occupational health in relation to physical and organizational ergonomics. Please, consults the following references: https://www.redalyc.org/pdf/2990/299023509007.pdf / https://www.jstor.org/stable/3100253 / https://www.psicothema.com/pdf/4206.pdf / https://www.scielo.cl/scielo.php?script=sci_arttext&pid=S0718-24492016000100004 . For such motivations, we have reason enough to use the EFA and then the CFA to validate our novel instrument in the same data. If there is a similar instrument or a priori solid theory in this regard in the specific context addressed by this research project, it would be recommended to use only CFA (but this is not our case). We understand that what the reviewer comments (using different data for EFA and CFA) is a typical procedure considering certain circumstances, such as the existence of a previous solid theory related to the instrument under validation. In this way, we consider that the writing of the manuscript does not clarify what is mentioned, therefore this confusion is generated.
In order to clarify the reviewer's concerns, the following lines were added or modified:
Lines 376-380, “…Table 11 shows the results taking into account the KMO – Bartlett test, where 13 observable variables were eliminated from the 31 original items, i.e., 31 items were analyzed for EFA …”
Lines 394-395, “…Once this part of the study is completed, the next step is to apply the Confirmatory Factor Analysis using the18 items that were the EFA results.”
Lines 424-430, “…Hence, from a total of 18 variables resulting (considering 31 items as inputs) from the Exploratory Factor Analysis, 6 of them are eliminated in the first test of the Confirmatory Factor Analysis for having low factor loadings of 0.3 and also for not complying with adequate adjustments considering the diverse goodness fit analysis. These were Q1_JS, Q3_JS, Q5_JS, Q14_EF, Q23_EO, and Q28_EO. In the second test of the Confirmatory Factor Analysis, were considering 12 variables as inputs, and then analyzed the Goodness Fit and the hypotheses to determine whether the model is accepted or rejected.”
Lines 559-573, “…An important aspect to clarify is that the sample size used in this project may appear to be small (i.e., this is a potential limitation). However, the sample size is relative to the model complexity, as well as to the a priori existence of strong theory related to the instrument to be validated [49-50]. In other words, in our case, there is no solid a priori theory or a similar instrument (which is a limitation), for this reason, the sample is small. In addition, said sample can be used for both the EFA and CFA, without losing reliability as part of the validation of our instrument. As future work, this proposed instrument should be further analyzed with other samples using only the CFA. Regarding the theoretical implications, our work presents evidence of an instrument with multiple ergonomic dimensions, which is not found in the literature concerning job satisfaction and occupational health. Also, regarding the practical implications, the results of this work can be used to promote and maintain a higher degree of physical, mental, and social well-being of workers as part of occupational health and job satisfaction.”
REVIEWER’S COMMENTS #7: Results and Discussion: The threshold loading value of 0.3 is very low; any reference? Where was CFA run? Where are the results?
RESPONSE TO REVIEWER #7: The comment is very valid to clarify the research design to the reader. Regarding the question, where was CFA run? the STATA 14 software was used in our manuscript. We add the following information: Lines 258-260, “…In particular, the STATA 14 software was used for descriptive data analysis and the implementation of different estimation techniques…”
Respect to the question, where are the results? For the intention of clarity, the CFA results can be presented either in tables or graphs. In our case, Figure 3 represents the result of the CFA of our work. Also, the CFA results are shown on lines 398-405, as well as in Table 13. If you wish, you can consult the following paper where the results of the CFA are presented graphically, https://doi.org/10.3390/rel12020079
Finally, regarding the comment, "...threshold loading value of 0.3 is very low; any reference?...", said value is recommended in various references. Please check the following reference:
Field, A. (2013) Discovering Statistics using SPSS, 4th edn. London: SAGE.
http://www.open-access.bcu.ac.uk/6076/1/__staff_shares_storage%20500mb_Library_ID112668_Stats%20Advisory_New%20Statistics%20Workshops_18ExploratoryFactorAnalysis_ExploratoryFactorAnalysis4.pdf
REVIEWER’S COMMENTS #8: Hypothesis Analysis: Since the values reported seem to be standardized regression coefficients, the interpretation of ‘…JS which explains 53% of the variance…’ is not correct
RESPONSE TO REVIEWER #8: Your comment is correct. We make some changes in the manuscript according to the appropriate technical words.
Lines 467-468, “…As a result, the EO has a direct positive relationship and impact with JS because the standardized regression coefficient is β = 0.53.”
Lines 483-485, “This means that Physical ergonomics has a direct positive relationship with Job Satisfaction, because the standardized regression coefficient is 0.37”
REVIEWER’S COMMENTS #9: There are some issues with the use of articles and commas., for example, the title reads… An Cross-Sectional…
RESPONSE TO REVIEWER #9: We made a typing error. The text was corrected.

Reviewer 3 Report
1. The abstract would begin with the knowledge gap.
2. The research design is not adequate. Structural equation modeling is proposed; however, the procedure does not explain how it was carried out, what software was used for data analysis.
3. 75 participants is too small a sample to use the multivariate technique of structural equation modeling.
4. The reporting of psychometric properties is deficient. For the validity of the instrument scores, you should specify whether the instrument was constructed by you or has been adapted from other contexts. If it was constructed by you, you would start with exploratory factor analysis to identify the dimensions of each construct. If it was an adapted instrument, then you could do validity based on internal structure using confirmatory factor analysis. It is not correct to do both factor analyses on the same sample. Each has a different purpose. For reliability analysis, this analysis is subsequent to the factor analysis where the items that did not have an adequate factor loading were eliminated.
5. After presenting the evidence of validity and reliability of the data obtained with the instruments, the analysis of the relationships is carried out by structural equation modeling. The fit indexes are reported to contrast whether the theoretical model has empirical support with the data obtained.
6. Due to methodological deficiencies, the results presented cannot be interpretable.
7. The writing should be improved: part of it is written in Spanish (lines 459-461).
Author Response
REVIEWER 3
REVIEWER’S COMMENTS #1: The abstract would begin with the knowledge gap.
RESPONSE TO REVIEWER #1: After analyzing the reviewer's comment, our team considered it pertinent to improve the structure and content of the abstract, specifying the contribution to the knowledge gap.
Please check out the following particular sentences:
“…In fact, there exists a knowledge gap about the relation between diverse ergonomics factors, job satisfaction, and occupational health, in the educational institution's context…”
“…As result, Cronbach's Alpha coefficient was determined, 0.9028, which is considered adequate to guarantee reliability (i.e., very highly magnitude). Therefore, after the Structural Equation Model, only 12 items present a strong correlation which has a good model fit of 0.036 based on the Root Mean Square Error of Approximation, 1.09 degrees of freedom for the Chi-square and 0.9 for Goodness of Fit Index, with a confidence level of 95 %...”
“…Where organizational, and physical factors have a positive impact on job satisfaction with a factor load of 0.37 and 0.53, respectively, with a p-value of 0.016 and 0.000, respectively.”
REVIEWER’S COMMENTS #2: The research design is not adequate. Structural equation modeling is proposed; however, the procedure does not explain how it was carried out, what software was used for data analysis.
RESPONSE TO REVIEWER #2: The comment is very valid to clarify the research design to the reader. In this way, the STATA 14 software was used in our manuscript. We add the following information: Lines 258-260, “…In particular, the STATA 14 software was used for descriptive data analysis and the implementation of different estimation techniques…”
REVIEWER’S COMMENTS #3: 75 participants is too small a sample to use the multivariate technique of structural equation modeling.
RESPONSE TO REVIEWER #3: Regarding the comment, “75 participants is too small a sample to use the multivariate technique of structural equation modeling”. Technically, the sample size used in SEM really depends on model complexity but also on many other factors (e.g., normality of the data, missing patterns, among others). That is, complex systems require larger samples than simple systems. Considering the above, regarding the sample size, various practical opinions and arguments exist in the literature. For example, Boomsma (1982), in “The robustness of LISREL against small sample sizes in factor analysis models” mentions that a minimum sample size of 100 or 200 is a good consideration. On the other hand, there are researches where sample sizes of 50-70 have been used, which have been sufficient to describe particular models. Please see the following reference: (https://doi.org/10.1177/0013164414525397)(https://www.tandfonline.com/doi/pdf/10.1207/s15327752jpa8501_02 ) Considering the above, we consider that a sample size of 75 is sufficient, in addition to considering that the population size is not large enough. Thus, in the conclusions section, arguments are provided to clarify the reviewer's concern. Lines 559-572, “…An important aspect to clarify is that the sample size used in this project may appear to be small (i.e., this is a potential limitation). However, the sample size is relative to the model complexity, as well as to the a priori existence of strong theory related to the instrument to be validated [49-50]. In other words, in our case, there is no solid a priori theory or a similar instrument (which is a limitation), for this reason, the sample is small. In addition, said sample can be used for both the EFA and CFA, without losing reliability as part of the validation of our instrument. As future work, this proposed instrument should be further analyzed with other samples using only the CFA. Regarding the theoretical implications, our work presents evidence of an instrument with multiple ergonomic dimensions, which is not found in the literature concerning job satisfaction and occupational health. Also, regarding the practical implications, the results of this work can be used to promote and maintain a higher degree of physical, mental, and social well-being of workers as part of occupational health and job satisfaction.”
REVIEWER’S COMMENTS #4: The reporting of psychometric properties is deficient. For the validity of the instrument scores, you should specify whether the instrument was constructed by you or has been adapted from other contexts. If it was constructed by you, you would start with exploratory factor analysis to identify the dimensions of each construct. If it was an adapted instrument, then you could do validity based on internal structure using confirmatory factor analysis. It is not correct to do both factor analyses on the same sample. Each has a different purpose. For reliability analysis, this analysis is subsequent to the factor analysis where the items that did not have an adequate factor loading were eliminated.
RESPONSE TO REVIEWER #4: The work team considered that the scales and instrument aspects are not clear. Therefore, to clarify this matter, the manuscript was improved. Please check Lines 281-288, “…Once the stratified samples were determined, the next step is to design our customized measuring instrument to determine job satisfaction and occupational health, which it is consists of closed-type questions, applying nominal coding levels for demographic variables and ordinal coding for the constructs of job satisfaction and the ergonomic aspects inherent of the operator and organizational process with a Likert scale of five response levels from descending to ascending scales: Totally dissatisfied (1), Dissatisfied (2), Neither satisfied nor dissatisfied (3), Satisfied (4), and Totally satisfied (5)…”
Respect to the comment, “…It is not correct to do both factor analyses on the same sample. Each has a different purpose. For reliability analysis, this analysis is subsequent to the factor analysis where the items that did not have an adequate factor loading were eliminated.” You are right. We added information to clarify your comment:
Lines 559-572, “…An important aspect to clarify is that the sample size used in this project may appear to be small (i.e., this is a potential limitation). However, the sample size is relative to the model complexity, as well as to the a priori existence of strong theory related to the instrument to be validated [49-50]. In other words, in our case, there is no solid a priori theory or a similar instrument (which is a limitation), for this reason, the sample is small. In addition, said sample can be used for both the EFA and CFA, without losing reliability as part of the validation of our instrument. As future work, this proposed instrument should be further analyzed with other samples using only the CFA. Regarding the theoretical implications, our work presents evidence of an instrument with multiple ergonomic dimensions, which is not found in the literature concerning job satisfaction and occupational health. Also, regarding the practical implications, the results of this work can be used to promote and maintain a higher degree of physical, mental, and social well-being of workers as part of occupational health and job satisfaction.”
REVIEWER’S COMMENTS #5: After presenting the evidence of validity and reliability of the data obtained with the instruments, the analysis of the relationships is carried out by structural equation modeling. The fit indexes are reported to contrast whether the theoretical model has empirical support with the data obtained. Due to methodological deficiencies, the results presented cannot be interpretable.
RESPONSE TO REVIEWER #5: Improvements were made regarding the structure of the document and its content, particularly related to the methodology. Evidence of these changes can be seen in the various comments answered in this document. Thus, we consider that the new wording, figures, tables, and analysis of results are clear and consistent.
REVIEWER’S COMMENTS #6: The writing should be improved: part of it is written in Spanish (lines 459-461).
RESPONSE TO REVIEWER #6: We made a mistake during the final format of the manuscript. We have already checked all the text in order to avoid the same error and improve the writing.

Reviewer 4 Report
In general the article, looking at the title, seems appropriate for the profile of the journal. However, the first thing I miss is the possible relevance of the covid period in the conclusions or results obtained. Within the text of the article there is no reference to or speculation about the possible influences of this fact.
I think it should be made clearer what the research question and/or objectives of the article are.
Similarly, it would be useful if the theoretical and/or practical implications for studies and those interested in the subject could draw appropriate conclusions.
I miss a self-criticism in which the limitations and difficulties of the research are highlighted. In the same way, it would be appropriate to propose future lines of research that could overcome the drawbacks cited and/or encountered.
I would have liked an explanation of how the hypotheses were deduced. That is to say, by virtue of what previous work and/or theories the possibility of formulating the hypotheses put forward has been deduced.
Last but not least, it would be necessary to explain the measurements of the scales. That is, have the scales used been validated by previous work? What are they? Cite them. If not, that they are original to the authors should be made clear.
I believe the methodology using structural equations is well applied and appropriate. It is the strong point of the article. Perhaps the authors could even consider sending it to other journals such as Mathematics from the same publisher (MDPI).
I recommend a revision of the English translation, especially because you can see some untranslated parts e.g. "In particular, changes made during the pandemic related to physical and organisational ergonomics affected job satisfaction (lines 459 and 460).
I hope I have contributed with my comments to the improvement of the publication,
Author Response
REVIEWER 4
REVIEWER’S COMMENTS #1: In general the article, looking at the title, seems appropriate for the profile of the journal. However, the first thing I miss is the possible relevance of the covid period in the conclusions or results obtained. Within the text of the article there is no reference to or speculation about the possible influences of this fact.
RESPONSE TO REVIEWER #1: After analyzing our manuscript, we confirm that various parts of the document refer to and speculate about the relevance of the pandemic on job satisfaction. in fact, references [39-41] support our argument. Please review the following parts of the document that support our answer:
Lines 224-227, “…Each of these factors can group the intercorrelated items that are, at the same time, relatively independent from others sets (factors) of items. In addition, another factor that has currently affected job satisfaction in all sectors of society and industry is related to the SARS-CoV-2 pandemic [39-41].”
Lines 246-249, “Also, it is important to mention that no studies were found that relate ergonomic factors in the three dimensions mentioned in the Structural Equation Model framework. Furthermore, there are no works related to job satisfaction research educational institutions using this model during the SARS-CoV-2 pandemic.”
Lines 523-542, “The above results allow us to infer that the work environment during the pandemic affected the job satisfaction of the interviewees. In particular, changes made during the pandemic related to physical and organizational ergonomics affected job satisfaction. Although a specific study related to cognitive ergonomics was not carried out, which studies the cognitive aspects of workers and the interaction with the work system, qualitatively, it was perceived that the level of stress of the jobs increased during the pandemic, which is related to the variables that most impact job satisfaction. In the same sense, the educational institution did not carry out a formal deployment of coping strategies against stress in workers, which was reflected in the results of job satisfaction.”
Also in the conclusions section, there is speculation about it, check out Lines 547-549, “With the development of this research, it was possible to know the parameters that have a close relationship with the dimensions already raised during the SARS-CoV-2 pandemic. Nowadays, there is a lack of studies about the dimensions of ergonomics regarding Job Satisfaction – Occupational Health and other variables…”
REVIEWER’S COMMENTS #2: I think it should be made clearer what the research question and/or objectives of the article are.
RESPONSE TO REVIEWER #2: After a review of the document, it was determined that the objective of the document is already present in lines 105-107, “Therefore, this study aims to determine the impact of ergonomic factors on job satisfaction and occupational health concerning unionized workers who provide operational services at this institution through structural equation analysis.”
REVIEWER’S COMMENTS #3: Similarly, it would be useful if the theoretical and/or practical implications for studies and those interested in the subject could draw appropriate conclusions.
RESPONSE TO REVIEWER #3: Subsection 5.3 (Impact on the Organizational Sustainability and implications) presents in detail various implications. For example, Lines 513-521: “…In the same sense, this study also presents very interesting practical implications. For example, the constant monitoring of the mentioned latent variables would allow knowing the job satisfaction level of particular periods and thus establish a job satisfaction baseline, which can serve to establish actions as countermeasures to improve job satisfaction level.” Even with the above, complementary information is added in the conclusions section. Please review the following:
Lines 559-572, “…An important aspect to clarify is that the sample size used in this project may appear to be small (i.e., this is a potential limitation). However, the sample size is relative to the model complexity, as well as to the a priori existence of strong theory related to the instrument to be validated [49-50]. In other words, in our case, there is no solid a priori theory or a similar instrument (which is a limitation), for this reason, the sample is small. In addition, said sample can be used for both the EFA and CFA, without losing reliability as part of the validation of our instrument. As future work, this proposed instrument should be further analyzed with other samples using only the CFA. Regarding the theoretical implications, our work presents evidence of an instrument with multiple ergonomic dimensions, which is not found in the literature concerning job satisfaction and occupational health. Also, regarding the practical implications, the results of this work can be used to promote and maintain a higher degree of physical, mental, and social well-being of workers as part of occupational health and job satisfaction.”
REVIEWER’S COMMENTS #4: I miss a self-criticism in which the limitations and difficulties of the research are highlighted. In the same way, it would be appropriate to propose future lines of research that could overcome the drawbacks cited and/or encountered.
RESPONSE TO REVIEWER #4: Subsection 5.3 (Impact on the Organizational Sustainability and implications), presents some limitations and difficulties. For example, Lines 517-540, “…Finally, this study presents some limitations that establish potential future research. One of the main aspects is the number of people considered in the study. This implies that it is not possible to increase the organizational sustainability level (considering a baseline) only by analyzing a particular group of personnel, that is, the study must be extended to more workers and from other departments.” Even with the above, complementary information is added in the conclusions section. Please review the following:
Lines 559-574, “…An important aspect to clarify is that the sample size used in this project may appear to be small (i.e., this is a potential limitation). However, the sample size is relative to the model complexity, as well as to the a priori existence of strong theory related to the instrument to be validated [49-50]. In other words, in our case, there is no solid a priori theory or a similar instrument (which is a limitation), for this reason, the sample is small. In addition, said sample can be used for both the EFA and CFA, without losing reliability as part of the validation of our instrument. As future work, this proposed instrument should be further analyzed with other samples using only the CFA. Regarding the theoretical implications, our work presents evidence of an instrument with multiple ergonomic dimensions, which is not found in the literature concerning job satisfaction and occupational health. Also, regarding the practical implications, the results of this work can be used to promote and maintain a higher degree of physical, mental, and social well-being of workers as part of occupational health and job satisfaction.”
- Izquierdo, I.; Olea, J.; Abad, F. J. Exploratory factor analysis in validation studies: Uses and recommendations. Psicothema 2014, 26, 395-400. https://doi.org/10.7334/psicothema2013.349
- Hurley, A. E.; Scandura, T. A.; Schriesheim, C. A.; Brannick, M. T.; Seers, A.; Vandenberg, R. J.; Larry, J. WilliamsExploratory and Confirmatory Factor Analysis: Guidelines, Issues, and Alternatives. J. Organ. Behav. 1997, 18, 667-683.
REVIEWER’S COMMENTS #5: I would have liked an explanation of how the hypotheses were deduced. That is to say, by virtue of what previous work and/or theories the possibility of formulating the hypotheses put forward has been deduced.
RESPONSE TO REVIEWER #5: The hypothesis presented (H1, H2, and H3) were deduced from the theory presented in subsection 2.1 (Dimensions of Ergonomics), and supported by references [16-25]. In fact, although all the references are important, perhaps the most relevant is: 23. Kohli, A.; Sharma, A. The Critical Dimensions of Job Satisfaction of Academicians: An Empirical Analysis. IUP J. Organ. Behav. 2018, 17, 21-35. Thus, as can be seen, the hypotheses are just after the mentioned information.
REVIEWER’S COMMENTS #6: Last but not least, it would be necessary to explain the measurements of the scales. That is, have the scales used been validated by previous work? What are they? Cite them. If not, that they are original to the authors should be made clear.
RESPONSE TO REVIEWER #6: The work team considered that the scales and instrument aspects are not clear. Therefore, to clarify this matter, the manuscript was improved. Please check Lines 281-288, “…Once the stratified samples were determined, the next step is to design our customized measuring instrument to determine job satisfaction and occupational health, which it is consists of closed-type questions, applying nominal coding levels for demographic variables and ordinal coding for the constructs of job satisfaction and the ergonomic aspects inherent of the operator and organizational process with a Likert scale of five response levels from descending to ascending scales: Totally dissatisfied (1), Dissatisfied (2), Neither satisfied nor dissatisfied (3), Satisfied (4), and Totally satisfied (5)…”
REVIEWER’S COMMENTS #7: I recommend a revision of the English translation, especially because you can see some untranslated parts e.g. "In particular, changes made during the pandemic related to physical and organizational ergonomics affected job satisfaction (lines 526-530).
RESPONSE TO REVIEWER #7: We made a mistake during the final format of the manuscript. We have already checked all the text in order to avoid the same error and improve the writing.

Round 2
Reviewer 2 Report
Dear Authors,
It is good to see the revised version of the article; this time, it is better.
Good Luck!
Reviewer 4 Report
I accept the changes and consider them appropriate